# Maternal Well-Being and Stage of Behaviour Change during Pregnancy: A Secondary Analysis of the PEARS Randomised Controlled Trial

**DOI:** 10.3390/ijerph20010034

**Published:** 2022-12-20

**Authors:** Doireann Roche, Anthony Rafferty, Sinead Holden, Sarah Louise Killeen, Maria Kennelly, Fionnuala M. McAuliffe

**Affiliations:** 1UCD Perinatal Research Centre, School of Medicine, University College Dublin, National Maternity Hospital, D02 YH21 Dublin, Ireland; 2UCD School of Mathematics and Statistics, University College Dublin, D04 V1W8 Dublin, Ireland

**Keywords:** well-being, behaviour change, lifestyle intervention, overweight, pregnancy, mHealth

## Abstract

We aimed to determine whether early pregnancy well-being was associated with the stage of behaviour change during an antenatal lifestyle intervention using a secondary analysis of data from the Pregnancy Exercise and Nutrition Research Study (PEARS). Pregnant women (*n* = 277) with well-being data in early pregnancy were included. Maternal well-being was measured using the World Health Organisation Five-Item Well-Being Index. The intervention consisted of a mobile health (mHealth) phone application, supported by antenatal education and exercise, to prevent gestational diabetes in a population with overweight. Stage of behaviour change was measured in late pregnancy using a five-stage classification. Ordinal logistic regression was used to examine if well-being, the study group, or their interaction, were related to behaviour change. Maternal well-being (OR 1.03, 95% CI 1.01, 1.04, *p* < 0.01) and the study group (OR 2.25, 95% CI 1.44, 3.51, *p* < 0.01) both significantly influenced the positive stage of behaviour change. The probability of being at stage 5 increased from 43 to 92% as well-being increased from 0 to 100% and was higher in the intervention (53%) compared to the control (34%) group (*p* ≤ 0.01 (8.65, 29.27). This study demonstrates the potential importance of well-being in enabling women to engage with a healthy lifestyle, and the role that mHealth technology has in facilitating beneficial behaviour change.

## 1. Introduction

Pregnancy can be a psychologically complex time during a woman’s life [1,2]. Previous research suggests that the antenatal period may be associated with increased levels of stress and anxiety, leaving women more vulnerable and susceptible to mental health difficulties [3,4]. The prevalence of depression and anxiety in pregnancy ranges from 4 to 25%, compared to only 5% in nonpregnant women of reproductive age [5,6,7]. This is thought to be even higher in women with a raised Body Mass Index (BMI), where a negative psychological state carries the potential to compromise both maternal and foetal health [8,9,10]. Individuals with pre-existing mental health concerns may be unable to balance the psychological, social, and physical demands of pregnancy with life’s daily challenges, thus creating a debilitating sense of inadequate ‘well-being’ [11].

There is no consensus on the definition for well-being, but at a minimum, it includes the absence of negative emotions, the presence of positive emotions, positive functioning leading to fulfilment, and satisfaction with one’s life [12,13]. Well-being encompasses both physical and mental health with the aim of promoting a more holistic approach to healthcare [14]. The World Health Organisation Five-Item Well-Being Index (WHO-5) is a self-reported assessment of subjective well-being [15]. It has been validated in screening for depression and for use as a measurable outcome of well-being in clinical research [16]. Well-being is thought to enable women to cope with the demands of pregnancy whilst satisfying their own needs [17]. In addition to affecting the maternal experience of pregnancy, well-being can influence pregnancy outcomes and subsequent childhood development [2,18]. Heightened levels of maternal stress and anxiety have been associated with reduced uterine blood flow and pregnancy-related health complications such as pregnancy-induced hypertension and preeclampsia [19,20]. Recent clinical practice guidelines in pregnancy recommend screening for mental health concerns [21]. Despite this, maternal well-being is not routinely evaluated during pregnancy, leading to missed opportunities to offer effective levels of support [22,23].

Along with well-being, lifestyle choices and maternal behaviour during pregnancy are known to have important implications for the unborn child [24]. Previous studies have demonstrated that suboptimal diet and smoking during pregnancy are associated with preterm birth, low birth weight, and neurodevelopmental disorders [25,26,27]. There is some evidence to suggest that women may be more willing to engage in health-promoting behaviours during pregnancy out of motivation for the good health of their child [28]. Therefore, pregnancy is thought to be a pivotal time when women are more receptive to making healthier lifestyle choices, yet many women grapple to do so [28]. A plethora of reasons have been suggested as to why this is so [29]. The antenatal period serves as the ideal opportunity for intervention when patients have regular contact with healthcare professionals. Despite this, many behavioural interventions trialled in the antenatal period fail to change the target behaviour [30].

There are a variety of theories in health psychology that help inform whether individuals change their behaviour [31]. In the stages of change model, an individuals’ “readiness to change” is classified in to one of five categories from “pre-contemplation (no intention or awareness of need to change) to “maintenance” (change has been made and the behaviour is being maintained) [31]. It is hypothesised that understanding the theory behind behaviour change may allow for tailored and more effective interventions, e.g. advice-giving, exploring ambivalence, or support [31]. This is of particular relevance in high-risk groups, such as those with obesity, as seen in this study, who would benefit the most from lifestyle and behaviour change [32]. The impact of women’s pre-existing state of well-being on stage of behaviour change in pregnancy is not well reported in the literature, particularly for women with overweight and obesity [33].

Access to additional support in pregnancy may enable women to engage in health-promoting lifestyle behaviours and have a better sense of personal well-being [34,35]. Traditionally, family, friends, general practitioners, and staff at maternity units were the predominant providers of social support in pregnancy [36,37]. However, during the COVID-19 pandemic, a global shift to more readily available online resources for antenatal support has arisen, which includes mobile phone technology [38,39]. The widespread use of smartphone technology, referred to as ‘Mobile Health’ or mHealth, is now increasingly used in healthcare settings [40]. New technologies have the potential to serve as an invaluable tool for managing chronic disease and promoting healthy behaviour without increasing the workload on clinicians [41,42,43,44,45]. They also have the potential to be cost-effective whilst giving women more autonomy over their health during pregnancy [40]. Therefore, it is prudent to assume that mHealth applications have the potential to support positive behavioural change and further impact maternal well-being through engagement with intervention [46].

The aim of our study was to investigate if maternal well-being in early pregnancy is associated with stage of behaviour change in women exposed to a novel healthy lifestyle smartphone application as part of a clinical randomised controlled trial. The hypothesis under investigation was improved maternal well-being is associated with being in a higher stage of behaviour change in pregnancy.

## 2. Materials and Methods

### 2.1. Study Design

This is a secondary analysis of data from the Pregnancy Exercise and Nutrition Research Study (PEARS) randomised controlled trial (RCT). The PEARS RCT included: an mHealth-supported, low-glycaemic index dietary advice and an exercise prescription among pregnant women with overweight or obesity. The original study was conducted at the National Maternity Hospital, Dublin, and results have previously been published [39,47]. The trial was registered with an international trial registry (ISRCTN registry, http://www.isrctn.com/ (accessed on 10 October 2021), ISRCTN29316280) and ethical approval was granted by the hospital ethics committee in October 2012. Written, informed consent was obtained from all participants prior to recruitment.

Women were recruited at their first antenatal visit between 2013 and 2016. Eligible women were those aged 18–45 years, with a singleton pregnancy between 10- and 18-weeks of gestation and had a BMI of >25 kg/m^2^ and <39.9 kg/m^2^. Exclusion criteria included a history of gestational diabetes mellitus (GDM) in antecedent pregnancies, any on-going medical disorder requiring treatment, multiple pregnancies, or no access to a mobile smartphone. Recruited participants were randomised to either the control or intervention group using a computer-generated randomisation system by a biostatistician (1:1). Participants were further stratified by BMI to ensure even numbers of participants in both groups. The randomisation was revealed at the participants’ first study visit. Participants received no compensation for their participation.

The intervention consisted of an antenatal ‘healthy lifestyle package’ led by a multidisciplinary team of nutritionists, midwives, and obstetricians. Patients in the intervention group were offered an initial education session during which they were given advice on exercise and plans for a low-glycaemic diet to follow during their pregnancy and access to the study-specific smartphone application. Women were encouraged to use the app daily. The app provided daily recipes, workouts, and a motivational quote of the day. Participants received emails every fortnight to track their compliance and had face-to-face meetings with the study team at 28 and 34 weeks. Women in the control group received standard antenatal care that did not include any standard advice around nutrition. The primary outcome of the PEARS trial was the incidence of diagnosed GDM at 28–30 weeks using the criteria set out by the International Association of Diabetes in Pregnancy Study Group [47,48]. While the intervention did not affect the diagnosis of GDM, it led to greater exercise participation and lower GI food intake [39,47]. A further secondary analysis from this trial exploring the effect of the intervention on the C3 complement and C-reactive protein has also been published elsewhere [49].

### 2.2. Study Population

The PEARS trial involved pregnant women (*n* = 565) with overweight or obesity, and the primary outcome was the incidence of GDM determined with oral-glucose-tolerance test values. For the current analysis, only women that completed the original trial, had a live born baby, and had documented WHO-5 Well-Being Index and stage of behaviour change scores in early (10–18 weeks of gestation) and late (28 weeks of gestation) pregnancy were included (*n* = 277).

### 2.3. Data Collection

All women had their baseline biometrics (height, weight, BMI) measured by a qualified professional in early pregnancy (initial visit, 10–18 weeks) and late pregnancy (28 weeks). Supplemental demographic information including: age, ethnicity, and level of education were recorded in early pregnancy. The Pobal Haase–Pratschke Deprivation Index address mapping tool was used to obtain neighbourhood deprivation data [50].

### 2.4. Maternal Well-Being

Maternal well-being was assessed using the World Health Organisation 5-Item Well-Being Index (WHO-5 Index) in early (10–18 weeks of gestation) and late pregnancy (28 weeks of gestation) [16]. This is a global tool that was previously validated in pregnancy [16]. Women were asked five questions about their well-being during the preceding two weeks. The questions assessed how frequently the woman felt (1) cheerful and in good spirits, (2) calm and relaxed, (3) active and vigorous, (4) fresh and rested, and (5) whether they felt their daily life had been filled with things that interest them. A Likert scale offered six possible answers, ranging from 0 to 5 points per question. The highest scoring response was “all of the time” (5 points), followed in descending order by “most of the time” (4 points), “more than half of the time” (3 points), “less than half of the time” (2 points), “some of the time” (1 point), and, finally, ‘at no time’ (0 points). The scores for each question were summed to give a final score ranging from 0 (lowest level of well-being) to 25 (highest level of well-being). A score < 13 was considered to be consistent with clinical depression [51]. To obtain a percentage score ranging from 0 to 100, the raw total score was multiplied by 4. A percentage score of 0 represented the worst possible score, whereas a percentage score of 100 indicated the highest level of well-being.

### 2.5. Behaviour Change

It was shown previously that women in the PEARS study did not differ in their preintervention stage of behaviour change. At baseline, most women were in the contemplation stage in both groups, while after the intervention, there was a significantly greater proportion of women in the maintenance phase [39]. As a result, the stage of behaviour change postintervention is the main outcome measure in this study. It was measured in late pregnancy (28 weeks of gestation) to assess women’s perceptions of their subjective readiness to engage in physical activity behaviours using a model previously validated in both pregnant and nonpregnant populations [52,53]. Women were determined by a self-reported questionnaire to be in one of five possible groups: precontemplation (stage 1), contemplation (stage 2), preparation (stage 3), action (stage 4), or maintenance (stage 5). This is based on the Transtheoretical Model or Stage of Change Model developed by Prochaska in the 1970s [54]. In the precontemplation stage, people do not plan to take action in the foreseeable future. In the contemplation stage, people start to recognize their behaviour may be problematic and intend to start engaging in healthier behaviours. In the preparation stage, people are ready to take action. In the action stage, people have changed their behaviour and plan to keep moving forward with healthy behaviour change. In the maintenance stage, people have sustained their behaviours for a period and intend to maintain them going forward.

### 2.6. Statistical Analysis

Ordinal logistic regression was used to determine if maternal well-being at baseline (initial recruitment visit, 10–18 weeks), study group allocation (smartphone app versus control), or the interaction between well-being and group allocation, influenced the stage of behaviour change postintervention (28 weeks of gestation). We examined the effect of well-being and study group allocation on the stage of behaviour change with and without an interaction term to determine if an interaction improved the model fit. For the model including the interaction term, the continuous well-being variable was centred prior to analysis to improve interpretation of the parameter estimates. The “margins” command was used to predict the probability of being at each stage of behaviour change per ten-unit increase in maternal well-being (0–100) and group allocation (smartphone app versus control). Prior to analysis, data were assessed for normality using Shapiro–Wilk tests and variables considered normally distributed are reported as mean ± standard deviation or median and interquartile range if found to have non-normal distribution. Data analysis was conducted using Stata statistical software (version 14.2) and an alpha of <0.05 was used to determine significance.

## 3. Results

Data from 277 women were analysed for this study and included 140 women allocated to the intervention group and 137 to the control group. Baseline maternal demographics are presented in Table 1. In summary, the population was predominantly white (90.6%) and had a mean age of 32.49 ± 4.35 years and a median BMI of 28.27 (26.73, 31.06) kg/m^2^s. Mean preintervention well-being in the study population was 57 ± 15%, with 30% of women having reduced well-being (<50%) and 3% having low well-being (<28%). Mean well-being scores preintervention were 58 ± 15% in the intervention group and 55 ± 15% in the control group (see Table 1).

### 3.1. Determinants of Behaviour-Change Stage

There was no significant interaction between well-being and group allocation variables (OR 1.02, 95% CI 0.99, 1.05, *p* = 0.17). The interaction term also did not significantly improve the model fit, so it was dropped from the model prior to the evaluation of the main effects. Maternal well-being (OR 1.03, 95% CI 1.01, 1.04, *p* < 0.01) and the study group (OR 2.25, 95% CI 1.44, 3.51, *p* < 0.01) both significantly influenced the stage of behaviour change postintervention.

### 3.2. Maternal Well-Being and Stage of Behaviour Change

Maternal well-being at each stage of behaviour change is shown in Table 2. For every ten-point increase in well-being, there was a corresponding shift in the probability of the stage of behaviour change (Figure 1). For stages 1–4, there was a decrease in the probability of being at each stage with every incremental increase in maternal well-being (decreasing from 4 to 0.5% at stage 1, 30 to 3% at stage 2, 16 to 3% at stage 3, and 7 to 2% at stage 4). The probability of being at stage 5 increased from 43 to 92% as maternal well-being increased from 0 to 100%.

### 3.3. The PEARS RCT Intervention and Behaviour Change

The predicted probability of being at behaviour-change stages 1, 2, or 3 was significantly higher in the control group postintervention. There was no difference in the predicted probability of behaviour-change stage 4 between the intervention and control groups (7% vs. 6%). The predicted probability of being at stage 5 postintervention was significantly higher in the intervention group postintervention (Table 3). The differences were greatest at behaviour-change stages 2 and 5, with women at stage 2 of behaviour change being 14 percentage points more likely to be in the control group (37% vs. 23%) and women at stage 5 being 19 percentage points more likely to be in the intervention group (53% vs. 34%).

## 4. Discussion

### 4.1. Principal Findings

We found that women with higher subjective levels of personal well-being in early pregnancy showed a higher tendency to change their health and lifestyle behaviours during pregnancy. Conversely, poor maternal well-being was significantly associated with the likelihood of behaviour change in pregnancy. Our study also showed that women were more likely to be in a positive stage of behaviour change when exposed to a healthy lifestyle intervention including a smartphone application.

### 4.2. Maternal Well-Being in Antenatal Care

Our findings highlight the potential significance of acknowledging maternal well-being as standard in maternity care. This is consistent with the recently published core outcome sets (COSs) for the prevention and treatment of postpartum haemorrhage, in which well-being is identified as an essential outcome for incorporation [55]. There can be significant variation in states of maternal well-being throughout pregnancy, when women are particularly vulnerable to states of poor overall health [56]. This is clearly reflected by the range of WHO-5 scores reported by women taking part in this trial. However, identifying those with poor well-being can be problematic, as often the women most at risk do not reach out and seek help [57]. Therefore, screening for maternal well-being is an opportunity for early detection and intervention centred around positive behaviour change [58,59].

Clinical practice guidelines have recently been updated to yield similar recommendations for mental health screening in pregnancy [21]. Professional bodies also recognise the risk of poor mental health and the need for clear pathways for the screening and management of perinatal mental health. The National Institute for Health and Care Excellence (NICE) clinical guidance recommends routine screening for mental health problems such as depression and anxiety [59]. While the WHO-5 Index was originally developed for assessing subjective well-being, recent evidence suggest that is has valid psychometric properties for assessing depressive symptoms [60,61,62]. A WHO-5 score of less than <28% has been correlated to a diagnosis of major depression using the DSM-IV criterion [63]. Here, we demonstrate the use of the validated WHO-5 Well-Being Index as a potential means of screening for maternal well-being in pregnancy. However, its use could potentially be expanded as a global screening tool for both perinatal mental health and well-being in everyday antenatal care.

With an effective strategy of screening for well-being in place in everyday antenatal care, we can identify women with low well-being earlier and considerably more consistently. With this information at our disposal from early on in pregnancy, we can offer earlier access to mental health services, increased contact with health professionals, and multidisciplinary involvement between obstetricians, psychiatrists, social workers, midwives, and general practitioners.

### 4.3. Stage of Behaviour Change and Well-Being

This study found that pregnancy is an opportunistic, ‘teachable moment’ for behaviour and lifestyle change [64]. Motivation to make a change is often driven by the awareness that the lifestyle choices a woman makes in pregnancy may have implications for both her and her unborn child [65]. Diet and exercise in pregnancy has been the focus of extensive research. Optimizing nutrition has the potential to improve outcomes for both the mother and the child [66]. Suboptimal diet, alcohol consumption, and smoking in pregnancy have been linked in previous studies to preterm birth, low birth weight, and neurodevelopmental disorders [25,26,67,68]. In addition, previous research suggests that by adopting healthy physical activity and dietary behaviours during pregnancy, women can reduce the risk of obstetric complications such as gestational diabetes and preeclampsia [69].

It is well reported in the literature that the motivation to make healthy changes in pregnancy is often driven by a desire to ensure the good health of the baby [70,71]. There are few longitudinal studies allowing the accurate interpretation of behaviour change in pregnancy and what factors influence a woman’s behaviour to change [65]. This study demonstrated that maternal well-being influenced a state of behaviour change postintervention. Women with a subjective sense of poor psychological well-being were less likely to change their behaviour in pregnancy. Our findings suggest that, by recognising women with poor well-being through screening, we can identify women who may be less likely to implement behaviour changes that are known to improve maternal and foetal outcomes. This provides a strategy to guide interventions in antenatal care [72].

### 4.4. Stages of Behaviour Change in Pregnancy

Here, we observed that women with overweight and obesity, for the most part, enter pregnancy wanting to make positive behaviour changes. Women with higher levels of personal well-being were more likely to transition from contemplation/preparation to maintenance when exposed to an mHealth-supported lifestyle intervention. If these behaviours are maintained, this has the potential to have long-lasting health benefits for both the mother and the child.

### 4.5. mHealth Intervention

With a global shift towards digital technology, including smartphone applications, there is increasing need to modernise routine pregnancy care [73]. Research has demonstrated that smartphone applications can improve readiness to engage with physical activity as well as dietary intake [74]. In the PEARS trial, the intervention group increased their moderate intensity exercise by on average 18 min more per week while the control group showed no difference. The PEARS trial equally succeeded in increasing adherence to healthy dietary advice in the intervention group compared to the control. Furthermore, this intervention employed several behaviour-change theories with the aim of effectively targeting behaviour constructs [48]. This allowed a systematic approach to evaluating the interventions effect on behaviour change in clinical practise [75]. A previous meta-analysis of interventions using behaviour-change theories highlighted their role in combatting low levels of physical activity in pregnancy [76]. Our findings suggest that behavioural change interventions, supported by mHealth, can work symbiotically to improve health-promoting behaviours amongst pregnant women.

Furthermore, mHealth is an exciting advancement in modern medicine with many advantages over traditional maternity care. It makes information readily available for pregnant women through smartphone applications, digital communications, or video consultations, all in the comfort of their own home [77]. In addition, it can reduce the problems caused by unequal access to healthcare in remote geographical locations, with long-distance health services referred to as telehealth [78]. Moreover, it has the potential to reduce long-term costs in maternity care by reducing the number of hospital visits, reducing transport costs for both patients and staff, and reducing time-to-diagnosis [77,79].

### 4.6. Strengths and Limitations

The key strengths of this study comprised the inclusion of novel data pertaining to well-being, behaviour change, and smartphone application support. Maternal well-being was also measured using a validated scoring system recommended by the World Health Organisation.

There are also some limitations worth noting, which include the secondary analysis of existing data, as well as the potential for social desirability bias in the intervention group, such that reported improvements in well-being and healthy behaviours may have been adjusted to echo the advice given to participants. Another limitation was the attrition rate in the data collection of the well-being and behaviour-change scores, which greatly reduced our study sample size compared to the original sample of the PEARS RCT. The nature of our sample makes it difficult to generalize results. Additionally, this study was exploratory in using existing data, meaning no sample size calculation was undertaken. Due to the relatively small sample, confounders were not included in this exploratory analysis. In future research, it would be informative to include other associated variables such as social support in the analysis. Finally, for the purpose of this RCT, the well-being index and stage of behaviour change scores were only recorded in early and late pregnancy. Future research is needed looking at the impact of the well-being index on the stage of behaviour change throughout all stages of pregnancy and postpartum to fully understand its short- and long-term implications on maternal and foetal health.

### 4.7. Clinical Implications

This study highlights the role of maternal well-being as a key predictor of women’s readiness to engage and change their behaviours in pregnancy. Adopting healthy behaviours could potentially lead to improved maternal and foetal outcomes in pregnancy. This study advocates for the need to routinely screen maternal well-being, opening up opportunities for early intervention to promote healthy lifestyle behaviours. The WHO-5 Well-Being Index is a free, easily accessible, standardised tool for assessing maternal well-being that is available in many languages and has been validated for use in pregnancy. The results of this investigation also support the adoption of mHealth, specifically smartphone apps, to complement traditional antenatal care to improve maternal lifestyle choices. The ‘Hollestic’ smartphone app is an example of one such app, launched at the National Maternity Hospital in Dublin and which is now available for free to download from the Google store, and which provides up-to-date information for nutrition and physical activity in pregnancy.

## 5. Conclusions

This study demonstrates the importance of maternal well-being in enabling women to engage in a healthy lifestyle, and the role that smartphone app technology has in facilitating beneficial behaviour change. Future research is needed to ascertain if these changes are maintained in a patient’s lifestyle postpartum. In addition, mHealth has the potential to transform healthcare through improved convenience, reduced costs, and demands on our healthcare system. However, its effectiveness and implementation will require ongoing evaluation from evidence-based information so that mHealth continues to develop and adapt to the current conditions of our healthcare system.

## Figures and Tables

**Figure 1 ijerph-20-00034-f001:**
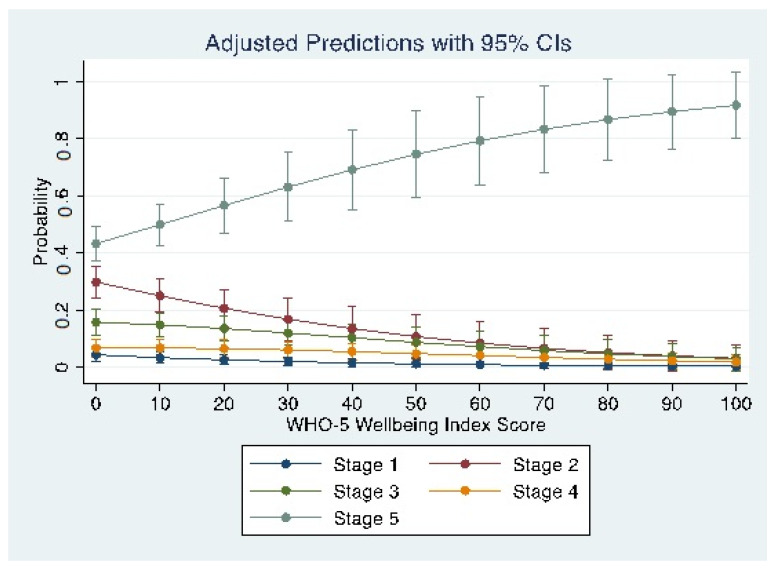
Predicted probabilities and 95% confidence intervals for each 10-point incremental increase in the percentage of maternal WHO-5 Well-Being Index scores for each stage of behaviour change.

**Table 1 ijerph-20-00034-t001:** Baseline characteristics of the study population (*n* = 277).

Baseline Characteristics	Intervention (*n* = 140)	Control (*n* = 137)
	Mean (SD)	Mean (SD)
Age (y)	32.60 (4.80)	31.76 (4.04)
Height (m)	1.64 (0.06)	1.65 (0.07)
Weight (Kg) ^†^	77 (70, 84)	78 (72, 85)
BMI (Kg/m^2^) ^†^	28 (27, 31)	28 (27, 31)
Gestation (w) ^†^	15 (14, 16)	15 (14, 16)
White *	132 (94)	122 (89)
Completed tertiary education *	83 (59)	107 (78)
Current smoker *	5 (4)	7 (5)
HP Pobal index Maternal well-being	6.33 (10.90) 55 (15)	6.13 (11.59) 58 (15)

* denotes *n* (%); ^†^ denotes non-normal data represented as median (interquartile range).

**Table 2 ijerph-20-00034-t002:** Baseline maternal well-being scores * at each stage of behaviour change. Values reported are mean percentage scores (standard deviation; SD).

	Maternal Well-Being Scores (Mean (SD))		
Stage of Behaviour Change	Overall (*n* = 277)	Intervention (*n* = 140)	Control (*n* = 137)
Stage 1 (precontemplation)	55 (11)	52 (5)	56 (4)
Stage 2 (contemplation)	53 (15)	55 (3)	52 (2)
Stage 3 (preparation)	55 (15)	52 (2)	57 (4)
Stage 4 (action)	56 (13)	59 (3)	54 (4)
Stage 5 (maintenance)	61 (14)	62 (2)	59 (2)
Overall Mean (SD)	57 (1)	58 (1)	56 (1)

* Maternal well-being scores were determined using the WHO-5 Well-Being Index.

**Table 3 ijerph-20-00034-t003:** Percentage predicted probabilities of postintervention behaviour-change scores based on study group allocation (intervention vs. control).

Stage of Behaviour Change	Intervention (*n* = 140)	Control (*n* = 137)	Percentage Difference	*p*-Value (CI)
Stage 1 (precontemplation)	3%	7%	−4%	<0.01 (−6.21, −1.06)
Stage 2 (contemplation)	23%	37%	−14%	<0.01 (−21.7, −6.11)
Stage 3 (preparation)	14%	16%	−2%	0.03 (−3.29, −0.14)
Stage 4 (action)	7%	6%	<1%	0.31 (−0.28, 0.86)
Stage 5 (maintenance)	53%	34%	+19%	<0.01 (8.65, 29.27)

## Data Availability

The data presented in this study are available upon request from the corresponding author.

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
