# Peer review of "Maternal Well-Being and Stage of Behaviour Change during Pregnancy: A Secondary Analysis of the PEARS Randomised Controlled Trial"

_ijerph, 2022, doi:10.3390/ijerph20010034_

Round 1

Reviewer 1 Report

The topic of wellness during pregnancy is of great interest, and if smart apps can be used to enhance positive health outcomes, it is an important area for exploration. The article is well written and promotes the integration of technology into health care, which relates well to the digital generation. There are two areas that could be improved: at the end of the literature review there is a brief mention that there is already a previous secondary analysis of the PEARS study (reference #51). There is no further mention of why this current study was undertaken if there is already a secondary analysis, or what gap was revealed by the review of that study (#51). This should be explained, to avoid suspicion of redundancy and more so, to identify the specific intentions of this research team to uncover further new data. Replication studies and re-analyses are common, however must be fully justified. In addition, in the discussion leading to the generic and broad conclusion, it would be nice to see further recommendations for how this research can be applied to healthy interventions or what other research should follow. This is an exciting area with much potential for health education and interventions, so some further ideas to continue research would be helpful. Overall, this is a meaningful article to promote maternal health using technology. 

Author Response

We appreciate the time and effort that you and the reviewers have dedicated to providing your valuable feedback on our manuscript. We are grateful to the reviewers for their insightful comments on our paper. We have been able to incorporate changes to reflect most of the suggestions provided by the reviewers. We have highlighted the changes within the manuscript.

Reviewer 2 Report

Dear Editors and authors,

I thank you for my consideration as a reviewer of this manuscript. I am pleased to contribute to International Journal of Environmental Research and Public Health. 

This research provides evidence of the importance of well-being for engaging in a healthy lifestyle, and the role of technology in pregnant women. I recommend considering some issues for the publication of the manuscript: 

The introduction section is well-justified with recent bibliographical citations. I would like to recommend including the hypothesis or research questions following the aim of this study. 

In the methodology section, if there is any compensation for the participants should be indicated in the manuscript. I suggest including evidence of the reliability of the instruments used, including for the sample of this research. 

In the discussion section, I would like to recommend that authors point out the potential benefits of the usemHealth because this kind of digital technology tool would reduce the long-term cost of health in maternity. Moreover, I suggest indicating the limitations and the difficulties for generalizing the results and including in the future analysis the role of other associated variables, such as social support. Finally, it could be interesting to examine the maintenance of these changes in lifestyle during the post-partum.

Author Response

(The authors gave the same response as above.)

Reviewer 3 Report

General comments

This paper presents an opportunity to analyse data from an RCT to address an important consideration in pregnancy – maternal wellbeing and healthy behaviour change. However, below I raise some concerns, particularly around the reporting and interpretation of results that I feel need to be addressed before publication

Detailed comments:

1. Choice and interpretation of outcome measure

- The first line of the abstract states “we aimed to determine whether early pregnancy well-being influenced behaviour change…” However, the authors later state that the primary outcome measure was motivation to change (Stage of Change), rather than behaviour change itself. I suggest that the authors should update the abstract and wording in other relevant sections to reflect this. (i.e. change to: “motivation to change behaviour”.

- It seems that behaviour (‘exercise participation’ and ‘lower GI food intake’) was recorded in the original PEARS study, could the authors explain why they chose to use Stages of Change, rather than actual behaviour change for this study?

2. Results

The authors state that an alpha of 0.05 was used to determine significance. This should be reported as < 0.05.

I question the interpretation of the results that the authors provide. The manuscript states (emphasis added): “These findings imply that the odds of being in a more advanced stage of behaviour change was 1 times greater for every unit increase in well-being and 2 times greater in the intervention group than the control.”

Please can the authors explain this interpretation based on the observed ORs?

For example, the OR for wellbeing is 1.03. To me this suggests a 3% increase in the odds of being in a more advanced stage of behaviour change. An odds ratio of 1 would indicate no difference between groups; similarly an increase of ‘1-times greater’ suggests equivalence between groups since 1x1=1, 2x1=2 etc.

Results section 3.1. is entitled ‘Determinants of behaviour change’; however as described above, this is misleading since the outcome reported here is Stage of Change. Please reword.

Results Section 3.2. Maternal wellbeing and behaviour change. The authors state that “For every ten-point increase in well-being, there was a corresponding shift in the probability of stage of behaviour change (Figure 1).”

While the Figure shows that an increased well-being score predicts Stage 5. The figure shows no clear impact of maternal wellbeing on the likelihood of being in the other Stages of Change. Error bars appear to overlap, and, contrary to what is reported in the text, the Figure appears to show that for low levels of wellbeing, participants were more likely to report Stage 2 or 3, than Stage 1.

Similarly, Table 2 does not show a consistent positive association between maternal wellbeing and Stage of Change, with the possible exception of increased scores at Level 5.

Results section 3.3. states “The predicted probability of being at behaviour change stages 1, 2 or 3 was significantly higher in the control group post intervention.” I suggest that the authors need to indicate the comparison group here e.g. higher in the control group vs the intervention group.

3. Discussion:

The authors should acknowledge and consider the impact of missing data on the results. The study sample was around 50% lower than the original sample from the PEARS RCT. This could introduce bias.

The discussion should be updated to reflect the correct interpretation of results, as described in the previous section.

The discussion does not make a clear case for how this evidence can be used to improve maternal wellbeing and behaviour change in practice. Although it discusses the importance of screening for wellbeing, it is not clear how this screening will inform care, other than offering women a healthy behaviour change intervention. For example, how should low well-being be addressed?

Author Response

(The authors gave the same response as above.)
